# Identification of a Monosomic Alien Chromosome Addition Line Responsible for the Purple Color Trait in Heading Chinese Cabbage

**Xiaoyun Xin** [1,2,3,4,†], **Deshuang Zhang** [1,2,3,4,†], **Hong Zhao** [1,2,3,4], **Tongbing Su** [1,2,3,4] , **Xiuyun Zhao** [1,2,3,4], **Weihong Wang** [1,2,3,4], **Peirong Li** [1,2,3,4], **Yangjun Yu** [1,2,3,4], **Jiao Wang** [1], **Shuancang Yu** [1,2,3,4,*] and **Fenglan Zhang** [1,2,3,4,*]

1. Beijing Vegetable Research Center (BVRC), Beijing Academy of Agriculture and Forestry Science (BAAFS), Beijing 100097, China
2. National Engineering Research Center for Vegetables, Beijing 100097, China
3. Key Laboratory of Biology and Genetic Improvement of Horticultural Crops (North China), Ministry of Agriculture, Beijing 100097, China
4. Beijing Key Laboratory of Vegetable Germplasm Improvement, Beijing 100097, China
* Correspondence: yushuancang@nercv.org (S.Y.); zhangfenglan@nercv.org (F.Z.); Tel.: +86-10-51503141 (S.Y.); +86-10-51503038 (F.Z.)
† These authors contributed equally to this work.

**Abstract:** Purple heading Chinese cabbage has become popular in recent years due to its attractive color and health benefits. However, purple varieties remain rare, and the regulation mechanism of anthocyanin accumulation in Chinese cabbage is still largely unknown. By introducing the purple color trait from *Brassica juncea*, a new purple heading Chinese cabbage cultivar (18M-245) was generated with deep purple leaves at both the seedling and adult stages. Anthocyanin accumulation in 18M-245 increased when grown at low temperatures. FISH and genotyping results showed that the purple trait was caused by an alien chromosome addition line derived from the *Brassica* B genome. The LDOX coding gene BjuB014115 from the addition line was highly expressed in 18M-245, consistent with the results of anthocyanin accumulation. Meanwhile, several MYB and bHLH transcriptional factors from the *Brassica* A genome were found to directly bind to the promoter of BjuB014115, suggesting that interactions between the *Brassica* A and B genomes are involved in the regulatory network of anthocyanin biosynthesis in Chinese cabbage. Our results provide new insights into the regulation mechanism of anthocyanin biosynthesis in purple heading Chinese cabbage.

**Keywords:** purple; anthocyanin; Chinese cabbage; addition line; *Brassica* genome variation; LDOX

## 1. Introduction

Chinese cabbage is an important traditional vegetable in Asian countries; its color is an important trait in terms of commercial value. The inner leaves of Chinese cabbage tend to be white; however, in recent years, new varieties with yellow and orange inner leaves at heading have been generated. Meanwhile, purple varieties remain rare, with coloration being associated with the accumulation of anthocyanins in the cell vacuole. Anthocyanin accumulation not only results in purple coloration, but also protects plants from biotic and abiotic stresses, as well as attracting insects for pollination [1]. Moreover, anthocyanins are also known to offer various health benefits as well as acting as a natural food dye [2].

Anthocyanins are the most prominent class of flavonoids and are widespread in nature. Accordingly, anthocyanin biosynthesis and the underlying regulatory mechanisms have been well documented in plants such as Arabidopsis, maize, petunia, apple, and several other commercial crops [3–5]. Anthocyanin biosynthesis can be divided into two steps: early and late [6,7]. The early pathway provides precursor substrates for flavonol and anthocyanin synthesis, and involves the structural genes *CHS* (*chalcone synthase*), *CHI* (*chalcone*

*isomerase*), *F3H* (*flavanone 3-hydroxylase*), *F3′H* (*flavanone 3′-hydroxylase*) and *FLS* (*flavonol synthase*), while the late pathway processes the synthesis of colored anthocyanin biosynthesis and subsequent modification, and involves the structural genes *DFR* (*dihydroflavonol 4-reductase*), *ANS* (*dihydroflavonol 4-reductase*)/*LDOX* (*leucoanthocyanidin dioxygenase*), UGT (*UDP-glucosyltransferase*) and *AT* (*acyltransferase*) [8,9].

The regulation of anthocyanin biosynthesis is mainly conducted at the transcriptional level via various MYB/bHLH/WDR transcription factors, which directly bind to a specific domain in the structural gene promoters, thereby affecting expression [10]. In Arabidopsis, the expression of EBGs (early biosynthesis gene) is induced by at least three R2R3-MYBs, namely PFG1/MYB12, PFG2/MYB11 and PFG3/MYB13 [11]. Meanwhile, the MYB-bHLH-WDR (MBW) ternany protein complex, which consists of an R2R3-MYB factor, R/B-like bHLH factor, and WD40 repeat factor, plays an important role in upregulating LBGs (late biosynthesis genes) [12–16]. In contrast, AtMYBL2 acts as an inhibitor of anthocyanin biosynthesis [17], repressing LBG expression by inhibiting the activity of the MBW complex [10,18].

Several purple genes have been mapped in *Brassica rapa*. For example, Hayashi et al. mapped the purple gene *Anp* in linkage group R07 of the previously reported *B. rapa* reference map using segregating F2 populations [19]. Meanwhile, Wang et al. mapped the purple gene *BrPur* in the A03 linkage group [20], while He et al. isolated the R3R3-MYB-regulating gene *BrMYB2* from the A07 chromosome, which is responsible for anthocyanin accumulation in purple Chinese cabbage 11S91 [21]. Furthermore, by conducting quantitative trait locus (QTL) analysis, Guo et al. identified a major QTL for anthocyanin content on the A09 chromosome, on which they subsequently identified two transcription factors (BrEGL3.1 and BrEGL3.2) as key regulator candidates in anthocyanin accumulation [22]. Similarly, Zhang et al. identified BrMYBL2.1 as a negative regulator of anthocyanin biosynthesis using QTL-seq analysis [23]. However, despite these findings, the underlying mechanism of anthocyanin accumulation in Chinese cabbage remains largely unknown.

Purple heading Chinese cabbage cultivars are mainly generated by introgression using green leaf Chinese cabbage as the recipient plant and naturally purple Brassicaceae plants as the donor. To date, green Chinese cabbage has been crossed with purple-leaf mustard [24], cai-tai (*Brassica compestris L.var.purpurea Bailey*) [25], turnip [19], and red cabbage [26] to generate the purple Chinese cabbage germplasm. Lee et al. developed the Reddish-purple color (RPCC, 2n = AA = 20) by crossing Chinese cabbage with red cabbage [26]. Transcription analysis has further revealed that anthocyanin accumulation is caused by a high expression of LBGs and the regulatory MYB genes BrMYB90, BrMYB75 and BrMYB2-1 [27]. Moreover, Xie et al. developed a purple color phenotype by crossing *Brassica rapa* Charming yellow with purple *Brassica juncea* Hunan Qianyang, then carried out RNA-seq analysis to reveal that a PAP1 homolog gene transferred from the *Brassica juncea* B genome was responsible for the anthocyanin over-accumulation phenotype [28].

Alien addition lines can facilitate studies of chromosome homology, gene location, gene expression, and gene cloning, as well as supporting the production of probes for genome painting following chromosome micro-dissection [29,30]. In this study, we aimed to create a new purple heading Chinese cabbage cultivar and further demonstrate the regulation mechanism of anthocyanin biosynthesis in Chinese cabbage. We introduced the purple color trait of purple mustard to the green heading Chinese cabbage Jiao Erye to generate a new purple heading Chinese cabbage cultivar 18M-245. Further fluorescence in situ hybridization (FISH), gene expression analysis and genotype analysis demonstrated that the highly expressing BrLDOX coding gene BjuB014115 from the *Brassica* B genome was responsible for the purple color trait of 18M-245. Furthermore, BjuB014115 was also a direct target of several MYB and bHLH transcriptional factors from the *Brassica* A genome. This study described a new purple Chinese cabbage heading cultivar and demonstrated that interactions of the *Brassica* A and B genomes are involved in the regulation of the mustard-derived purple color trait in Chinese cabbage.

## 2. Materials and Methods

### 2.1. Plant Materials

The purple F1 hybrid 15NG28 was generated by crossing pak choi (*B. rapa*, AA, 2n = 20) with Chinese mustard (*B. juncea*, AABB, 2n = 36) as described in Figure S1 and previously [28]. The green Chinese cabbage parent Jiao Erye, a local Chinese cabbage, represents an inbred line self-crossed for more than 12 generations. Jiao Erye was also used as the green Chinese cabbage control in the analysis of anthocyanin content and gene expression. An F2 population consisting of 99 individuals was developed from a single purple-colored F1 individual obtained from the cross between 18M-245 and Jiao Erye.

### 2.2. Growth Conditions

Chinese cabbage seeds were germinated at 25 °C on 1/2 MS medium in the dark for 36 h, then transferred to soil and grown at 22 °C under a photoperiod of 16 h light/8 h dark. For low-temperature treatment, 2-week-old seedlings were grown at 4, 12 and 22 °C for 4 weeks, respectively.

### 2.3. Analysis of Anthocyanin Content

2.3.1. Analysis of Anthocyanin Content by Colorimetric Method

The anthocyanin content measurement was performed as Teng et al. previously described, with slight adjustments [31,32]. Six-week-old Chinese cabbage leaves grown at 4, 12 and 22 °C for 4 weeks, respectively, were collected and weighed, then ground into a powder using liquid nitrogen and incubated in 1 mL 2% hydrochloric acid in methanol at 4 °C for 4 h. The mixture was centrifuged at $10,000 \times g$ for 30 min at 4 °C, then the supernatant was transferred to a clean test tube and brought to a constant volume of 1 mL with 2% hydrochloric acid in methanol. Absorbance was measured at 530, 620 and 650 nm using a spectrophotometer (L7, INESA Scientific Instrument Co., Ltd, Shanghai, China). At least three individual plants per line were sampled, then anthocyanin content was calculated as follows:

$$OD_\lambda = (OD_{530} - OD_{620}) - 0.1*(OD_{650} - OD_{620})$$

$$\text{Anthocyanin content (nmol/g)} = OD_\lambda/4.60*106*FW(g)*1,000,000$$

2.3.2. HPLC–MS/MS Analyses

Outer and inner leaves of 12-week-old 18M-245 and parent green Chinese cabbage Jiao Erye plants were collected from three biological repeats per cultivar. Lyophilized leaf samples (100 mg) were then ground into a powder using TGrinder (OSE-Y50, TIANGEN, Beijing, China) and mixed with 1 mL extraction buffer (0.1% hydrochloric acid, 50% formic acid in water). They were then vortexed for 5 min and sonicated for 5 min before carrying out centrifugation at 12,000 rpm/min for 3 min at 4 °C. The supernatant was then filtered through a 0.22 μm millipore filter (Millex-GP, Darmstadt, Germany).

Anthocyanin compounds were analyzed using a C18 column (1.7 μm, 2.1 × 100 mm; ACQUITY BEH, Waters, Shanghai, China) with chromatographic separation using a binary gradient consisting of solution A (0.1% formic acid in water) and solution B (0.1% formic acid in methyl alcohol) at a flow rate of 0.35 mL/min and a column temperature of 4 °C. Solution B was introduced at concentrations of 5–50% from 0.00–6.00 min, 50–95% from 6.00–12.00 min, 95% from 12.00–14.00 min, and 5% from 14.00–16.00 min. Quantification was performed at 520 nm.

Isolated anthocyanin compounds were further identified using tandem mass spectrometry (MS/MS) (QTRAP® 6500+, SCIEX, Framingham, MA, USA) with electrospray ionization (ESI) as follows: positive ion model, 550 °C desolvation, 5500 V capillary voltage, and 35 psi curtain gas.

### 2.4. Fluorescence In Situ Hybridization (FISH)

#### 2.4.1. Chromosome Preparation

Young flower buds (approximately 1–2 mm long) were harvested from 40 back-crossed lines grown in a greenhouse, then immediately fixed in freshly prepared Carnoy's solution (100% ethanol: glacial acetic acid = 3:1) for at least 4 h at 4 °C. Fixed flower buds were then transferred to 70% ethanol and stored at 4 °C until use. Anthers were dissected from the treated flower buds, then soaked in distilled water for 30 min at room temperature before incubating in 2.5% cellulose and 2% pectolyase at 37 °C for 2.5 h. They were then soaked in distilled water for 30 min, transferred to a clean glass slide, then macerated in ethanol/acetic acid (3:1, *v/v*) using fine-point forceps. After air-drying, the slides were examined using an Axio Imager 2 (Zeiss, Jena, Germany) phase-contrast microscope and suitable preparations were stored at −20 °C until use.

#### 2.4.2. DNA Probe Preparation

The *Brassica* B genome was isolated from leaves of 4-week-old black mustard seedlings. *Brassica* A genome marker-derived CenBr1 and CenBr2 plasmids were donated by the Beijing Vegetable Research Center (BVRC). All DNA was labeled using Dig or Bio Nick Translation Mix (Roche, Basel, Switzerland) according to the manufacturer's instructions.

#### 2.4.3. FISH

The in situ hybridization protocol was carried out as previously described [33], with some modifications. Briefly, slides were washed for 5 min in 2 × SSC with Tween 20 at room temperature, then washed again in 2 × SSC without Tween 20. The chromosomes were denatured in 70% deionized formaldehyde in 2 × SSC buffer for 10 min at room temperature, then dehydrated in 75, 85 and 100% ethanol for 5 min each at –20 °C. The hybridization mixture (30 μL per slide) contained 50 ng of labeled CenBr1, 100 ng labeled CenBr2 or 100 ng labeled *Brassica* B genome DNA per slide; 50% formamide; and 10% dextran sulfate in 2 × SSC and salmon sperm blocking DNA at 100-fold excess of the labeled probes. The mixture was denatured at 95 °C for 10 min, chilled on ice for at least 10 min, then added to each slide. Chromosome preparations and pre-denatured probes were denatured at 80 °C for 3 min, then hybridized overnight in a humid chamber at 37 °C. Post-hybridization washes were then carried out for 15 min in 2 × SSC at room temperature. Slides were blocked for 40 min at 37 °C with 5% (*w/v*) BSA in 4 × SSC plus 0.2% Tween 20. Hybridization sites of digoxigenin- and biotin-labeled probes were then detected using anti-digoxigenin-rhodamine (11207750910, Roche, Basel, Switzerland) and avidin-fluorescein (A-2001-5, Vector Laboratories, Newark, California, United States), respectively. The chromosomes were counterstained with 1.5 μg/mL 4,6-diamidino-2-phenylindole (DAPI) mounted in Vectashield antifade (H-1200-10, Vector Laboratories, Newark, CA, USA).

#### 2.4.4. Image Processing

Slides were examined under an Axio Imager 2 microscope (Zeiss, Jena, Germany) equipped with epifluorescence illumination and filter sets for DAPI, FITC and Texas-Red fluorescence. At least 30 metaphase samples were examined for hybridization. Selected images showing blue, red and green fluorescence were acquired separately and processed using Zen2 software (https://www.zeiss.com.cn/microscopy/products/microscope-software/zen-2-starter.html, accessed on 3 April 2020). Global image adjustments for contrast, brightness, and color saturation were carried out using Photoshop software (v.CS5).

### 2.5. Real-Time Quantitative PCR (RT-qPCR)

Healthy leaves of Jiao Erye and 18M-245 plants grown at 4, 12 and 22 °C for 4 weeks were collected, then total RNA was extracted using an RNAprep pure plant kit (DP441, TIANGEN, Beijing, China). First-strand cDNA was synthesized using a Prime ScriptTM reagent Kit with a gDNA Eraser (RR047A, Takara, Dalian, China). RT-qPCR was performed using SYBR Green PCR master mix (04887352001, Roche, Basel, Switzerland) and a Light-Cycler 480 Real-Time PCR system (Roche, Basel, Switzerland). GADPH was used as an internal control. Primer sequences are listed in Table S1.

### 2.6. Genotype Identification

The *LDOX* coding sequence of 18M-245 was amplified by PCR using cDNA as the template. The coding sequence was inserted into the pEASY-Blunt simple cloning vector (CB111, TransGen, Beijing, China) then the recombined vector was sequenced using M13F and M13R primers. Primers for genotype identification were designed based on genomic sequence differences between Bra013652 and BjuB014115 as shown in Figure S1B. Total DNA extracted from leaves of 4-week-old seedlings using the CTAB method was used as a template for genotyping. Three different bands were obtained following PCR amplification: a 1906 bp A1 band and 1618 bp A2 band, which corresponded to *Bra013652*, and a 1360 bp A3 band corresponding to *BjuB014115*. Sequences of primers and bands are listed in Table S1.

### 2.7. Yeast-One Hybridization

Genomic DNA of 18M-245 was isolated from healthy leaves using the CTAB method. The *cis*-element in the 1.5-kb DNA sequence upstream of the start codon ATG of BjuB014115 was analyzed using the Plantcare online tool (http://bioinformatics.psb.ugent.be/webtools/plantcare/html/, accessed on 30 July 2019). The −516 to −427 bp region in the BjuB014115 promoter was cloned using 18M-245 genomic DNA as a template, then inserted into the yeast-one hybrid bait vector pAbAi. Coding sequences (CDSs) of BrEGL3, BrTT8, BrTTG1, BrMYBL2-1 and BrMYBL2-2 were cloned using 18M-245 cDNA as a template, then inserted into the prey pGADT7 vector. The pAbAi-BjuB014115 plasmid was then co-transformed with respective recombinant pGADT7 vectors into *Saccharomyces cerevisiae* strain Y1HGold and screened using SD/-Leu medium at 30 °C for three days. The pAbAi-BjuB014115 plasmid was also co-transformed with the empty pGADT7 to detect potential auto-activation. Interaction between the transcriptional factors and promoter region was then identified by transferring the positive clones to SD/-Leu medium with 800 ng/mL ABA.

## 3. Results

### 3.1. Generation of a New Purple Chinese Cabbage Germplasm

As previously reported, the F$_1$ hybrid 15NG28 was obtained by crossing green pak choi with Chinese mustard using a natural hybridization method [34]. Here, the 15NG28 hybrid was crossed with the inbred green Chinese cabbage Jiao Erye, then purple BC1 plants were selected and self-crossed twice (Figure S1). The new purple Chinese cabbage germplasm 18M-245 displayed the same traits as 15NG28, with a deep purple color on the adaxial and abaxial surfaces of the leaves, veins and adaxial petioles at both the seedling and adult stages (Figure 1A–C). A deep purple color was also displayed in both the outer and inner leaves of the leaf head (Figure 1D).

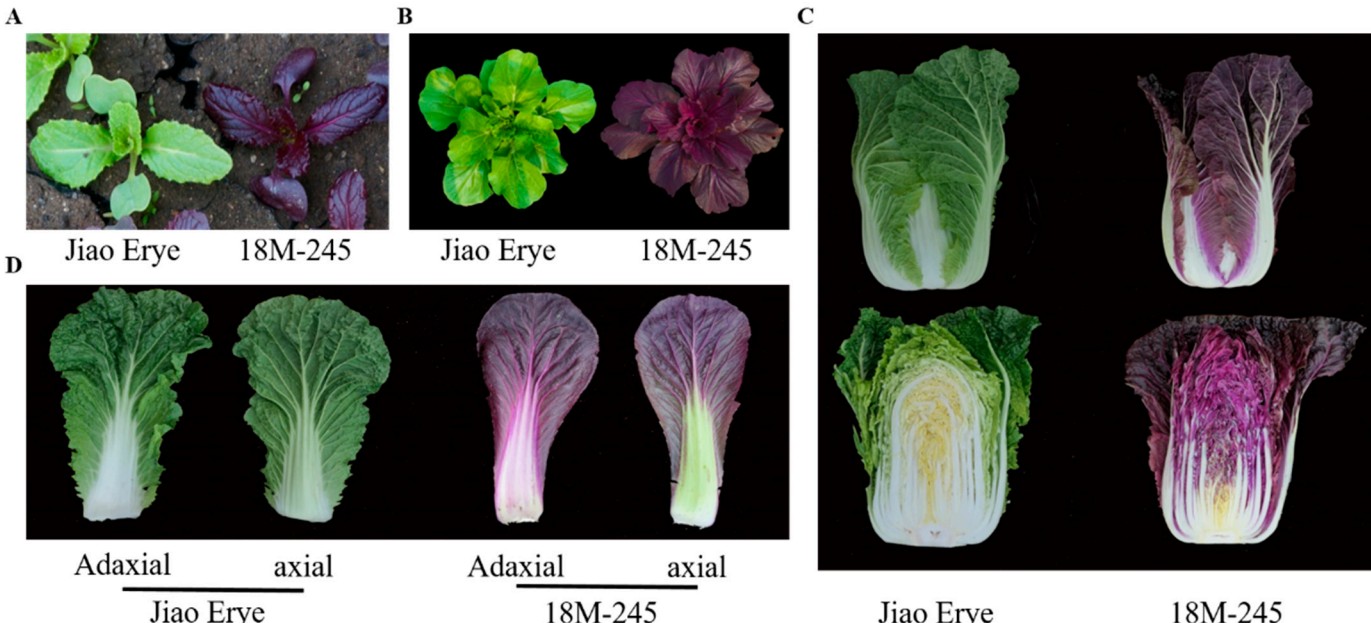

**Figure 1.** The purple color trait of 18M-245. Compared with the green Chinese cabbage parent, 18M-245 displayed deep purple leaves at both the seedling (**A**) and adult stages (**B,C**). A deep purple color was also displayed in both the outer and inner leaves of the leaf head (**C**), and on adaxial and abaxial surfaces of the leaves, veins and adaxial petioles (**D**).

### 3.2. Anthocyanin Content in the 18M-245 Leaves

Purple coloration in plant organs is usually caused by anthocyanin accumulation. The anthocyanin content of both green and purple Chinese cabbage leaves was therefore determined under normal conditions (22 °C). As shown in Figure 2A,B, leaves of the green Chinese cabbage Jiao Erye contained almost no anthocyanin under the above growth conditions, while those of 18M-245 contained 2.0 μmol/g·FW anthocyanin. Anthocyanin compounds in 18M-245 and Jiao Erye leaves were subsequently determined based on the HPLC–MS/MS method (Table 1). A total of 37 anthocyanin compounds were identified, of which cyanidin was the major component in both the purple and green Chinese cabbage, accounting for 96.77 and 98.02%, respectively (Table 1). Compared to the green parent, nine cyanidins, five pelargonidins, six peonidins, two delphinidins, one petunidin and one malvidin were significantly upregulated in 18M-245 (Table 1). The accumulation of cyanidin-3,5,3′-O-triglucoside and cyanidin-3-O-(6-O-p-coumaroyl)-glucoside was the main cause of the purple coloration in 18M-245, accounting for 17.8 and 5.9 μg/g DW, respectively (Table 1). Meanwhile, three flavonoid compounds were also detected in 18M-25. Compared to the green parent, Kaempferol-3-O-rutinoside was also significantly upregulated in 18M-245, accounting for 18.12 μg/g DW (Table 1).

Low temperatures were previously found to induce anthocyanin accumulation in purple heading Chinese cabbage and apple [35,36]. We therefore investigated anthocyanin accumulation in 18M-245 after low-temperature treatment. As shown in Figure 2B, growth for four weeks at 12 °C significantly increased the anthocyanin content of the purple Chinese cabbage leaves by about 100% compared to 22 °C, while no significant differences were observed in the green Chinese cabbage. The anthocyanin content of purple Chinese cabbage was also significantly increased compared to 22 °C. However, no further increases were observed at 4 °C compared to 12 °C. These results suggest that temperatures below 12 °C induce an increase in anthocyanin accumulation in purple Chinese cabbage.

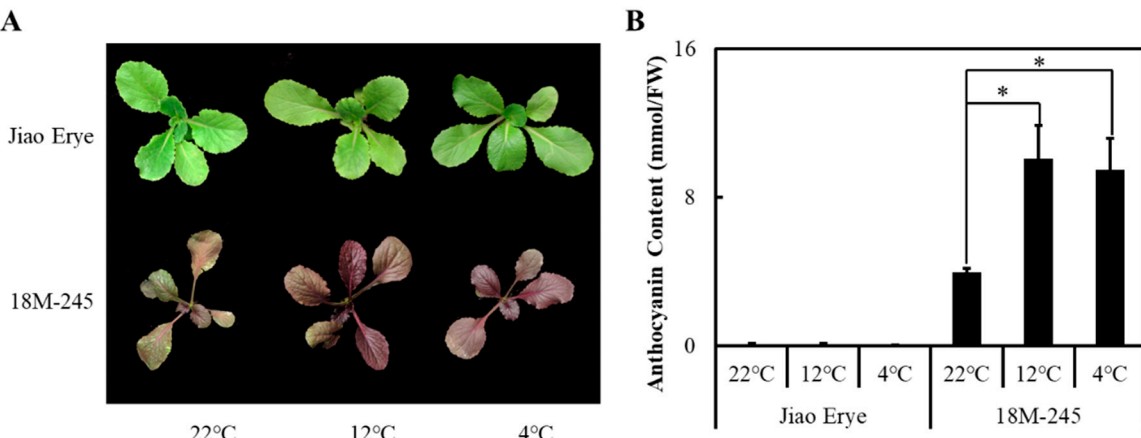

**Figure 2.** Anthocyanin accumulation in 18M-245 under low-temperature conditions. Phenotypes of Jiao Erye and 8M-245 grown at 22, 12 and 4 °C for 4 weeks, respectively (**A**). Leaf anthocyanin contents of four-week-old Jiao Erye and 18M-245 leaves at each temperature (**B**). Data are shown as the mean value ± SE. Asterisks indicate significant differences; Student's *t*-test (* $p < 0.05$).

### 3.3. Identification of a Monosomic Alien Chromosome Addition Line in 18M-245

To determine the inheritance pattern of the purple color trait in 18M-245, we generated an F2 population by crossing 18M-245 with its green parent Jiao Erye. Separation of the green and purple leaf color occurred in the F2 individuals without any intermediate color types, suggesting that the purple color trait is qualitative. A total of 84 out of the 99 F2 individuals displayed a purple leaf color, which did not fit the expected ratio for a single dominant allele. We therefore predicted that the purple leaf phenotype is not controlled by a single gene. FISH experiments were therefore carried out to investigate variation in the purple Chinese cabbage genome. As shown in Figure 3, 18M-245 contains 21 chromosomes. Using *Brassica*-A-genome-specific chromosome markers, 20 of these 21 chromosomes were found to display blue and red fluorescence, suggesting the existence of an addition line that does not belong to the *Brassica* A genome. Since the purple color trait of 18M-245 was derived from Chinese mustard, which contains both *Brassica* A and B genomes, we therefore carried out a second FISH experiment using *Brassica*-B-genome-specific markers. As a result, only the monosome that showed no blue or red fluorescence displayed green fluorescence. These results suggest that 18M-245 contains a monosomic alien addition line derived from the *Brassica juncea* B genome.

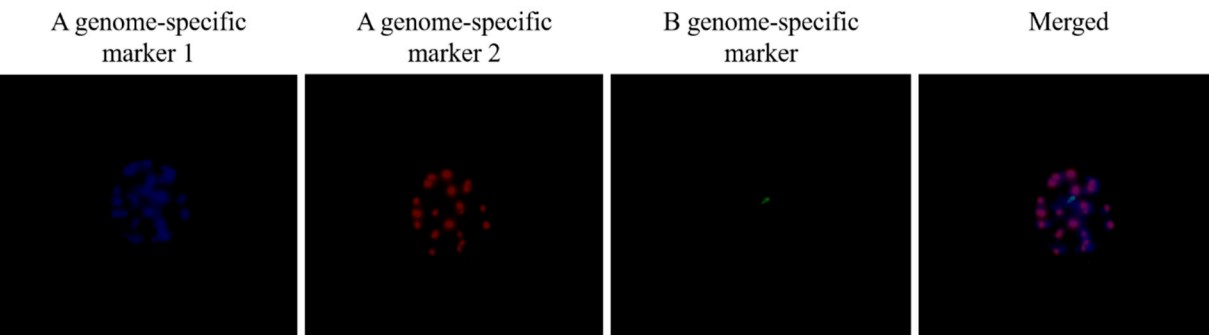

**Figure 3.** 18M-245 contains a monosomic chromosome derived from the *Brassica* B genome. FISH was performed using yellow flower buds from 18M-245. *Brassica*-A-genome-specific chromosomes were marked using A-genome-specific markers revealing blue and red fluorescence, while only one monosomic chromosome was marked with the B-genome-specific marker revealing green fluorescence, suggesting that the addition line was derived from the B genome.

### 3.4. LDOX Transferred from the Brassica B Genome Is Highly Expressed in 18M-245

Previous RNA-seq analysis has shown that the anthocyanin-synthesis-related structural genes *BrLDOX* and *BrDFR* are significantly induced in purple Chinese cabbage [28]. To determine the gene response of anthocyanin accumulation, we therefore used RT-qPCR to examine the expression of *BrLDOX* and *BrDFR* in Jiao Erye and 18M-245 grown at 22, 12 and 4 °C for four weeks. As shown in Figure 4A, the expressions of *BrLDOX* and *BrDFR* was upregulated in 18M-245 by ~1000 and ~10,000 fold compared to its green parent Jiao Erye at 22 °C, respectively. At 12 °C, *BrLDOX* expression increased significantly by about 20% compared to 22 °C. Meanwhile, no significant increase in *BrLDOX* expression was observed at 4 °C compared to 12 °C, consistent with the results of anthocyanin accumulation. Meanwhile, the expression of *BrDFR* showed no significant difference among the three different temperatures. The expressions of other structural genes in anthocyanin biosynthesis were also determined in Jiao Erye and 18M-245 grown at different temperatures (Figure S3). The expression of the most genes (except FLS2, which is depressed at 4 °C) was obviously induced in 18M-245 at 4 °C, inconsistent with the anthocyanin accumulation pattern of 18M-245 at low temperatures (Figure S3). These results suggest that the upregulation of *BrLDOX* is related to anthocyanin accumulation in 18M-245.

BrLDOX is coded by Bra013652 in the *Brassica* A genome. *BjuB014115* on the *Brassica* B05 chromosome is homologous to Bra013652, with the CDS of BjuB014115 sharing 94.06% similarity with Bra013652 (Figure S2A). To determine whether the increased *LDOX* expression in 18M-245 was caused by the monosomic addition line derived from the B genome, we therefore cloned and sequenced the *BrLDOX* coding sequence from 18M-245. As shown in Figure S2A, the coding sequence shared 100% similarity with BjuB014115, suggesting that the upregulation of *BrLDOX* expression in 18M-245 was caused by a homologous gene on the *Brassica* B05 chromosome.

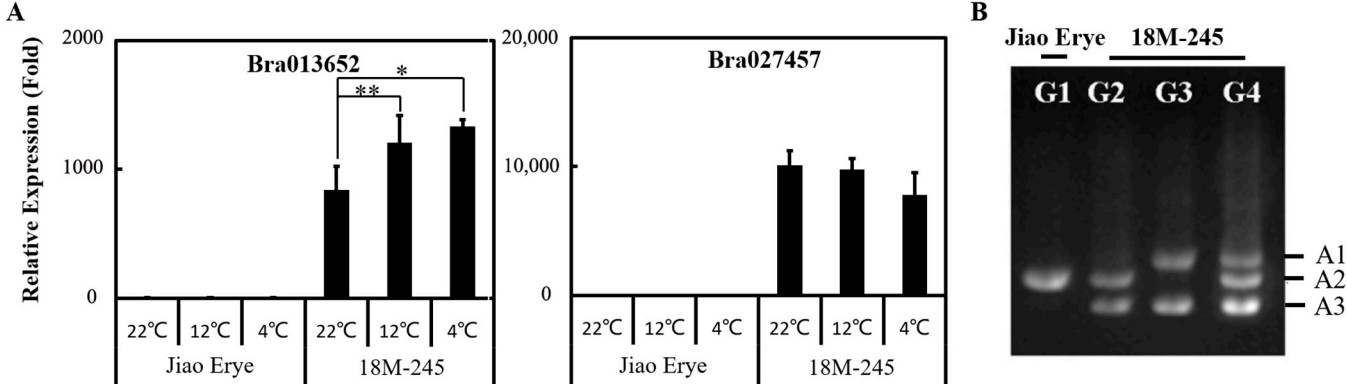

**Figure 4.** *BrLDOX* transferred from the B genome is highly expressed in 18M-245. Expression of anthocyanin-related structural genes *BrLDOX* and *BrDFR* at different temperatures was analyzed using RT-qPCR (**A**). Expression of *BrLDOX* was induced, while *BrDFR* expression decreased after low-temperature treatment. *GADPH* was used as internal control. Genotypes of green and purple Chinese cabbage (**B**). The 2016 bp A1 band and 1605 bp A2 band correspond to the Bra013652 gene sequence, while the 1306 bp A3 band corresponds to the BjuB014115 gene sequence. Data are shown as the mean value ± SE. At least three biological repeats of each RT-qPCR experiment were performed. Asterisks indicate significant differences; Student's *t*-test (* $p < 0.05$, ** $p < 0.01$).

### 3.5. BjuB014115 Is Necessary for the Purple Color Trait in 18M-245

To determine whether BjuB014115 is necessary for the purple color trait in 18M-245, we designed genotype-identification primers based on the genomic sequence differences between *Bra013652* and *BjuB014115* (Figure S2B). As shown in Figure 4B, we obtained three different bands and four different genotypes in Jiao Erye and 18M-245. After sequencing, bands A1 and A2 were found to correspond to the *Bra013652* gene sequence, while band A3 corresponded to the *BjuB014115* gene sequence, which was displayed in

all purple genotypes. The F2 population generated by crossing 18M-245 with the green Chinese cabbage parent Jiao Erye was subsequently genotyped. As shown in Figure 5, all 84 purple individuals displayed the A3 band corresponding to BjuB014115, while the 15 green individuals displayed only the A2 band corresponding to Bra013652. These results suggest that the purple color phenotype in 18M-245 is tightly associated with the *LDOX* gene derived from the *Brassica* B05 chromosome.

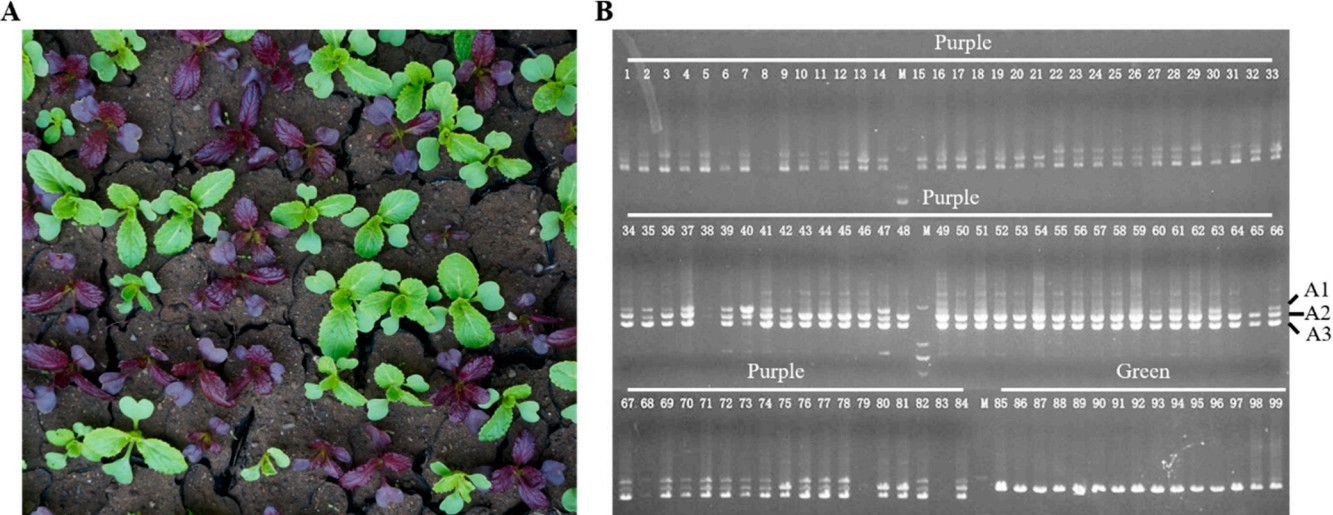

**Figure 5.** BjuB014115 is necessary for the purple color trait in 18M-245. The leaf color phenotype of the F2 separating population obtained by backcrossing 18M-245 with its green Chinese cabbage parent Jiao Eyra (**A**). Genotypes of the F2 population. A total of 84 purple leaf individuals (Nos. 1–84) and 15 green leaf individuals (Nos. 85–99) were used for genotype identification. The 1036 bp A3 band corresponds to BjuB014115 and was displayed in the purple individuals only, suggesting that the purple trait is tightly associated with the monosomic chromosome addition line derived from B05 (**B**).

*3.6. BjuB014115 Is Regulated by MYB and bHLH Transcription Factors from the Brassica A Genome*

In Arabidopsis, the expression of *AtLDOX* is regulated by several transcription factors, including R2R3-MYB transcription factors such as PAP1, PAP2, MYB113, MYB114, and MYBL2, and bHLH transcription factors such as EGL3 and TT8 [9]. Previous transcriptome analysis has revealed that *BrEGL3* (Br027796, B genome homolog BjuB028220), *BrTT8* (Bra027796, B genome homolog BjuB004115), *BrTTG1* (Bra009770, B genome homolog BjuB038062), *BrMYBL2-1* (Bra007957, B genome homolog BjuB047637) and *BrMBL2-2* (Bra016164, B genome homolog BjuB047637) are significantly upregulated in purple Chinese cabbage (Table 2), suggesting that these genes may also play a role in the regulation of anthocyanin biosynthesis in 18M-245. We therefore cloned and sequenced the coding sequences of these five transcription factors from 18M-245. The sequence homology analysis showed that the coding sequences of all five transcription factors were transcribed from the Brassica A genome (Figure S4).

Transcription factors regulate the expression of their target genes through interactions with specific *cis*-regulatory DNA elements usually located in the promoters. It is widely accepted that the MBW complexes including R2R3-MYB and basic helix–loop–helix (bHLH) transcription factors together with WD-repeat proteins play an important role in the transcriptional regulation of anthocyanin biosynthesis [9,37–39]. The structural genes DFR, LDOX, BAN, TT12, TT19 and AHA10 have been determined to be the direct targets of MBW complexes in *Arabidopsis*, and the regulatory module in the promoter of these structural genes contains at least an MYB-binding site, an AC-rich element and a bHLH-binding site [9]. Moreover, yeast-one hybrid and electrophoretic mobility shift analyses showed that MdMYB6 inhibits anthocyanin in apple through direct binding to the MYB-binding site in

the promoters of MdANS and MdGSTF12 [40]. We therefore analyzed the *cis*-element in the 1.5 kb DNA sequence upstream of the start codon ATG. Accordingly, three bHLH-binding sites (G-boxes) and two MYB-binding sites were identified within the −516 to −427 bp region (Figure 6A). This region was therefore cloned and inserted into the yeast-one hybrid bait vector pAbAi and co-transformed with pGADT7 recombinant plasmids transformed with respective coding sequences of the five upregulated transcription factors from 18M-245. The resulting positive clones were then screened on SD/-Leu medium with 800 ng/mL ABA. As shown in Figure 6B, all five positive clones grew well on the selective medium, and after 10-fold dilution, positive clones containing pGADT7-BrTT8, pGADT7-BrEGL3 and pGADT7-BrMYBL2-2 continued to grow slowly. These results indicate that BrTT8, BrEGL3, BrTTG1 and BrMYBL2 can bind to the *cis*-elements of the BjuB014115 promoter, suggesting that the expression of BjuB014115 is regulated by MYB and bHLH transcription factors from the *Brassica* A genome.

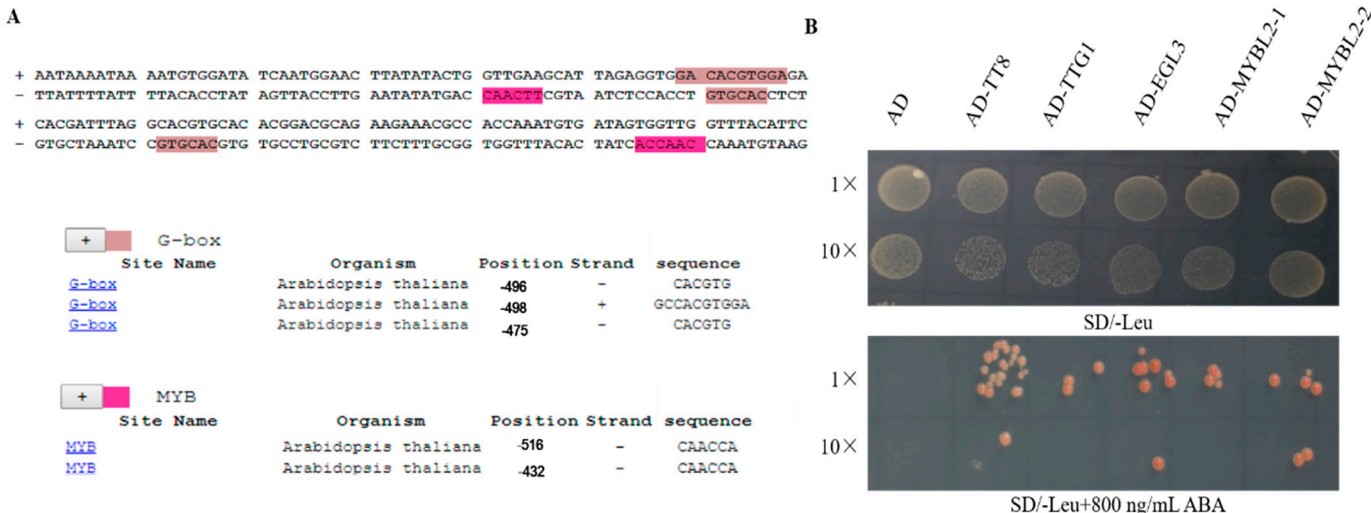

**Figure 6.** BjuB014115 is regulated by MYB and bHLH transcription factors from the Brassica A genome. Distribution of bHLH-binding site G-boxes and MYB-binding sites in the promoter of BjuB014115 (**A**). Interactions between TFs from the *Brassica* A genome with promoter fragments of BjuB014115 were examined using yeast-one hybridization (**B**). Positive clones were screened in SD/-Leu medium, then interactions between the TFs with the BjuB014115 promoter fragment were identified in 800 ng/mL ABA. All clones contained the recombinant AD plasmid. Positive clones were then diluted 10-fold with sterile water to further test their resistance to ABA.

**Table 1.** The contents of different anthocyanin compounds in the 18M-245 and Jiao Erye leaves based on HPLC–MS/MS.

| Class | Compound | Content (ng/gDW) | | Fold Change (18M-245 vs. Jiao Erye) | *p* Value |
|---|---|---|---|---|---|
| | | Jiao Erye | 18M-245 | | |
| Cyanidin | Cyanidin-3-O-glucoside | 31.88 | 26.39 | 0.83 | 0.702 |
| Cyanidin | Cyanidin-3,5,3′-O-triglucoside | 0.01 | 17.81 | 1191.97 | 0.005 |
| Cyanidin | Cyanidin-3-O-(6-O-p-coumaroyl)-glucoside | 0.00 | 5.95 | | 0.148 |
| Cyanidin | Cyanidin-3-O-sophoroside | 0.00 | 3.29 | | 0.003 |
| Cyanidin | Cyanidin-3,5-O-diglucoside | 0.00 | 2.47 | | 0.001 |
| Cyanidin | Cyanidin-3-(6′′-caffeylsophoroside)-5-glucoside | 0.00 | 1.19 | | 0.002 |
| Cyanidin | Cyanidin-3-O-5-O-(6-O-coumaroyl)-diglucoside | 0.00 | 0.99 | | 0.019 |
| Cyanidin | Cyanidin-3-(6-O-p-caffeoyl)-glucoside | 0.70 | 0.81 | | 0.817 |
| Cyanidin | Cyanidin-3-O-sambubioside-5-O-glucoside | 0.00 | 0.03 | | 0.022 |
| Cyanidin | Cyanidin-3-O-xyloside | 0.00 | 0.02 | | 0.048 |
| Cyanidin | Cyanidin-3-O-(6′′-ferulylsophoroside)-5-glucoside | 0.01 | 0.01 | 1.16 | 0.922 |
| Cyanidin | Cyanidin-3-O-sambubioside | 0.00 | 0.01 | | 0.012 |

**Table 1.** *Cont.*

| Class | Compound | Content (ng/gDW) | | Fold Change (18M-245 vs. Jiao Erye) | *p* Value |
|---|---|---|---|---|---|
| | | Jiao Erye | 18M-245 | | |
| Pelargonidin | Pelargonidin-3-sophoroside-5-glucoside | 0.00 | 0.05 | | 0.009 |
| Pelargonidin | Pelargonidin-3-O-galactoside | 0.29 | 0.04 | 0.14 | 0.167 |
| Pelargonidin | Pelargonidin-3-O-glucoside | 0.00 | 0.03 | | 0.007 |
| Pelargonidin | Pelargonidin-3-O-sophoroside | 0.00 | 0.03 | | 0.010 |
| Pelargonidin | Pelargonidin-3-O-sophoroside-5-O-(malonyl)-glucoside | 0.00 | 0.02 | | 0.040 |
| Pelargonidin | Pelargonidin-3-O-(6-O-p-coumaroyl)-glucoside | 0.00 | 0.01 | | 0.324 |
| Pelargonidin | Pelargonidin-3-O-rutinoside | 0.01 | 0.00 | 0.15 | 0.165 |
| Peonidin | Peonidin-3,5-O-diglucoside | 0.01 | 0.06 | 4.44 | 0.018 |
| Peonidin | Peonidin-3-O-glucoside | 0.00 | 0.05 | | 0.013 |
| Peonidin | Peonidin-3-sophoroside-5-glucoside | 0.00 | 0.05 | | 0.023 |
| Peonidin | Peonidin-3-O-5-O-(6-O-coumaroyl)-diglucoside | 0.00 | 0.02 | | 0.023 |
| Peonidin | Peonidin-3-O-(6-O-p-coumaroyl)-glucoside | 0.00 | 0.01 | | 0.015 |
| Peonidin | Peonidin-3-(caffeoyl-glucosyl-glucoside)-5-glucoside | 0.00 | 0.00 | | 0.044 |
| Delphinidin | Delphinidin-3-O-sophoroside | 0.66 | 0.54 | 0.82 | 0.832 |
| Delphinidin | Delphinidin-3,5-O-diglucoside | 0.08 | 0.12 | 1.41 | 0.649 |
| Delphinidin | Delphinidin-3-O-(6-O-malonyl-beta-D-glucoside) | 0.00 | 0.04 | 8.78 | 0.088 |
| Delphinidin | Delphinidin-3-O-sambubioside | 0.00 | 0.01 | | 0.016 |
| Petunidin | Petunidin-3-O-(6-O-malonyl-beta-D-glucoside) | 0.03 | 0.04 | 1.29 | 0.763 |
| Petunidin | Petunidin-3-O-glucoside | 0.00 | 0.01 | | 0.218 |
| Petunidin | Petunidin-3-O-sambubioside | 0.00 | 0.00 | 0.00 | |
| Malvidin | Malvidin-3-O-(6-O-malonyl-beta-D-glucoside) | 0.00 | 0.05 | Inf | 0.012 |
| Malvidin | Malvidin | 0.00 | 0.00 | 0.23 | 0.525 |
| flavonoid | Kaempferol-3-O-rutinoside | 0.00 | 18.12 | | 0.020 |
| flavonoid | Quercetin-3-O-glucoside | 5.63 | 2.23 | 0.40 | 0.539 |
| flavonoid | Naringenin-7-O-glucoside | 0.21 | 0.13 | 0.65 | 0.760 |

**Table 2.** Expression data of MYB and bHLH transcription factors in RNA-seq data as reported previously [34].

| Gene ID | Gene Name | Green Reads Count | Purple Reads Count | log2Ratio (Green/Purple) | *p*-Value | FDR |
|---|---|---|---|---|---|---|
| Bra027796 | BrEGL3 | 90 | 176 | −1.06819 | 0.457398 | 1 |
| Bra037887 | BrTTG8 | 3 | 4408 | −10.5618 | $6.57 \times 10^{-7}$ | 0.00207 |
| Bra009770 | BtTTG1 | 528 | 1069 | −1.11916 | 0.427705 | 1 |
| Bra007957 | BrMYBL2-1 | 4 | 463 | −6.911 | 0.000222 | 0.038438 |
| Bra016164 | BrMYBL2-2 | 45 | 2875 | −6.0951 | 0.000392 | 0.055515 |

## 4. Discussion

Chromosome engineering can be used to transfer beneficial genes to cultivated species in order to genetically improve them. Accordingly, genes from cabbage can be transferred to Chinese cabbage due to the close relationship between the two [41]. The purple color trait in Chinese cabbage is mainly derived from purple-colored Brassica plants via anthocyanin accumulation, which usually results from the transfer of structural or regulatory genes from red cabbage, purple mustard, purple flowering *Brassica rapa*, and so on. Zhang et al. developed a purple Chinese cabbage via an interspecies cross between Chinese cabbage and purple-leaf mustard [42], while a genetic analysis revealed that the purple color trait was caused by the insertion of fragments from the purple mustard A02 chromosome. Meanwhile, an analysis of these fragments revealed that they contained twelve structural and regulatory genes, of which five structural genes were highly expressed.

In this study, we generated a new purple Chinese cabbage germplasm (18M-245) and, using FISH and genotype analysis, demonstrated that the purple color trait was controlled by an addition line derived from the B genome of *Brassica juncea*. The *BrLDOX* coding gene BjuB014115 was also highly expressed in 18M-245. Among all the structural genes tested, only the expression of BjuB014115 was consistent with the anthocyanin accumulation

pattern in 18M-245 at low temperatures, suggesting that the expression change of *BrLDOX* is the main cause of anthocyanin accumulation in 18M-245. These results suggest that highly expressed structural genes transferred from the *Brassica* B genome are responsible for the Chinese cabbage purple color trait derived from purple mustard.

Anthocyanin accumulation not only gives plants an attractive color, but also improves their tolerance to environmental stresses. Gould reported that anthocyanin plays a primary role in tolerance to stressors as diverse as drought, cold, ultraviolet (UV)-B, and heavy metals, as well as in defense against herbivores and pathogens [43]. This is due to its role as an antioxidant in protecting the plant from biotic and abiotic stresses by scavenging ROS [44]. It has been reported that anthocyanin accumulation can be promoted in purple head Chinese cabbage seedling and apple fruit [35,36]. Jiang et al. reported that MdMYB2 in apple was able to respond to cold stress and enhance the corresponding stress tolerance by promoting anthocyanin accumulation [45]. In this study, the anthocyanin content in six-week-old 18M-245 leaves was significantly induced at 12 °C and 4 °C compared to 22 °C, which is similar to the previous report [36]. Thus, we predicted that the increased accumulation of anthocyanin in 18M-245 might assist Chinese cabbage in defense to cold stress.

It is widely accepted that the purple leaf is a qualitative character in Chinese cabbage, since no intermediate leaf color types have been identified in individuals derived from an interspecies cross between green Chinese cabbage and purple mustard. However, the major control genes and mechanism of anthocyanin biosynthesis in purple Chinese cabbage remain largely unknown. A homolog of the MYB transcription factor PAP1 transferred from the B genome of *Brassica juncea* was previously identified as being responsible for the purple color trait in Chinese cabbage [28]. Furthermore, the *Brassica*-B-genome-derived structural gene *CHS* and transcription factor MYBL2 were found to be highly expressed in purple Chinese cabbage [23]. Despite this, the regulatory network of anthocyanin biosynthesis in purple Chinese cabbage has yet to be determined. In this study, the BrLDOX coding gene BjuB014115 was highly expressed in 18M-245. Moreover, several upregulated MYB and bHLH factors including BrTT8, BrEGL3, BrTTG1 and BrMYBL2 were found to directly bind to the promoter of BjuB014115. The bHLH proteins BrTT8, BrEGL3 and the WD-repeat protein BrTTG1 are known to be involved in the MBW complexes. Different MBW complexes control anthocyanin biosynthesis in a tissue-specific manner [8,46]. In vegetative tissues, the anthocyanin biosynthesis pathway is promoted by PAP1/MYB75, PAP2/MYB90, MYB113 and MYB114 together with the combined action of a bHLH protein such as GLABRA3 (GL3), ENHANCER OF GLABRA3 (EGL3), or TT8, and the WD-repeat protein TTG1 [47], which is consistent with the anthocyanin accumulation traits of 18M-245 leaves. Meanwhile, AtMYBL2 is known as a repressor of anthocyanin biosynthesis in Arabidopsis. AtMYBL2 bound directly to TT8 and this complex represses the expression of DFR and TT8 [18], indicating that there may be antagonistic regulatory pathways of anthocyanin biosynthesis in 18M-245. Unlike the previous study, all of these highly expressed transcription factors in 18M-245 were coded by the *Brassica* A genome, suggesting that the interaction between *Brassica* A genome genes and structural genes transferred from the *Brassica* B genome is involved in the regulation of anthocyanin biosynthesis in purple Chinese cabbage derived from purple mustard. Taken together, our findings describe a new purple Chinese cabbage cultivar 18M-245 with a monosomic alien chromosome addition line derived from the *Brassica* B genome, and demonstrate that the highly expressing *BrLDOX* transferred from the *Brassica* B genome is a direct target of several transcription factors from the *Brassica* A genome, providing new insights into the anthocyanin biosynthesis regulation mechanism and breeding of purple heading Chinese cabbage.

**Supplementary Materials:** The following supporting information can be downloaded at: https://www.mdpi.com/article/10.3390/horticulturae9020146/s1, Figure S1: Generation of new purple Chinese cabbage 18M-245; Figure S2: The nuclear acid sequence alignment of Bra013652 and BjuB014115; Figure S3: Expression pattern of structural genes under three different temperatures; Figure S4: Homology tree showing the similarity of TFs CDS cloned from 18M-245 with the homologous genes from Brassica A and B genome; Table S1: Primers used in this study.

**Author Contributions:** Conceptualization, X.X. and D.Z.; methodology, X.X. and H.Z.; software, X.X., H.Z. and J.W.; validation, D.Z., X.X., X.Z. and W.W.; formal analysis, X.X., T.S., Y.Y. and P.L.; investigation, T.S.; resources, T.S. and S.Y.; data curation, X.X.; writing—original draft preparation, X.X. and D.Z.; writing—review and editing, S.Y. and T.S.; visualization, X.X.; supervision, S.Y. and F.Z.; project administration, T.S.; funding acquisition, D.Z. and X.X. All authors have read and agreed to the published version of the manuscript.

**Funding:** National Natural Science Foundation of China, grant number 31872094; Beijing Joint Research Program for Germplasm Innovation and New Variety Breeding, grant number G20220628003-01; Foundation for Young Scientists of BAAFS, grant number QNJJ202241; Innovation Project of Beijing Vegetable Genebank Resource Breeding and Optimization (KJCX20200113).

**Data Availability Statement:** The data that supports the findings of this study are available from the corresponding author upon reasonable request.

**Acknowledgments:** *Brassica*-A-genome-specific marker 1 and marker 2 used in the FISH experiment were kindly donated by Guixiang Wang from Beijing Vegetable Research Center (BVRC).

**Conflicts of Interest:** The authors declare no conflict of interest.

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
