# Peer review of "Identification of a Monosomic Alien Chromosome Addition Line Responsible for the Purple Color Trait in Heading Chinese Cabbage"

_horticulturae, doi:10.3390/horticulturae9020146_

Round 1

Reviewer 1 Report

The ms is interesting and in general well written. Only minor details of English language should be revised.

Abstract should be greatly imporved in order to fit the general structure. Introduction should be reduced and results should be discussed in a deep manner.

Some specific comments are included in the document.

Author Response

Dear reviewer:

Thank you very much for giving us the opportunity to revise this manuscript. We appreciate your constructive and insightful comments to improve our manuscript. Below, we address all the comments and questions point-by-point. And of course the text has been changed accordingly. But we failed to upload the revised manuscript here. We will email the editor later to find a proper way to send the revised manuscript to you. 

Best wishes!

Reviewer 2 Report

A minor revision is suggested in the introduction. Pl rephrase the final paragraph of the introduction to make the objectives clearer. 

Parts of results that are included in the introduction (lines 89 to 96) can be shifted to results and discussion sections.

Author Response

Dear reviewer:

Thank you very much for giving us the opportunity to revise this manuscript. We appreciate your constructive comments to improve our manuscript. Below, we address all the comments and questions point-by-point. And of course the text has been changed accordingly, which can also be tracked in the attached revised manuscript.

Comment 1: A minor revision is suggested in the introduction. Pl rephrase the final paragraph of the introduction to make the objectives clearer.

Parts of results that are included in the introduction (lines 89 to 96) can be shifted to results and discussion sections.

Response:Thanks for the reviewer’s insightful comments. We reworded the final paragraph of the introduction. Description about the aim of the study was added and the redundant description of results was deleted.

Reviewer 3 Report

Thank you for giving us the opportunity to consider your work.

Overall, this is a clear, concise, and well-written manuscript. The experimental results obtained are good, but the quality of the pictures is not all good.

Requires minor revision, so the authors must complete all the notes mentioned below

Introduction

Lines 44-50: You must provide the reference.

Materials and Methods

Line 113-123: Can a reference be added to this analysis of anthocyanin content?

Results

Describe the major findings of your study in the opening sentence.

Lines 324-325:  This sentence "materials and methods" ?

Figure 5. B : The picture is not clear

All results should be clearly described and logically arranged. References to the respective tables or figures should be in brackets after the statement, e.g., (Table 1) or (Fig. 3A). Use subheadings only when the text is too long and complicated

Discussion

The discussion needs to be visited. Appropriate comparisons of results obtained with those found previously should be presented, so I suggest this discussion be re-worded.

Line 403-405:  Can you expand and justify why this is the case?

Line 417-422: Extend the conclusion by writing main results of this study. The conclusion has to be improved.

References

Please check all references as several in the main text are old. The author must new cite references at least 75% of the latest reviews (e.g., from 2017-2022).

Author Response

Dear reviewer:

Thank you very much for giving us the opportunity to revise this manuscript. We appreciate your constructive comments to improve our manuscript. Below, we address all the comments and questions point-by-point. And of course the text has been changed accordingly, which can also be tracked in the attached revised manuscript.

Thank you for giving us the opportunity to consider your work.

Overall, this is a clear, concise, and well-written manuscript. The experimental results obtained are good, but the quality of the pictures is not all good.

Requires minor revision, so the authors must complete all the notes mentioned below

 Introduction

Comment 1: Lines 44-50: You must provide the reference.

Response: Thanks for the reminding. We added reference to these sentences.

Materials and Methods

Comment 2: Line 113-123: Can a reference be added to this analysis of anthocyanin content?

 Response: Thanks for the reminding. The anthocyanin content measurement was performed as Teng et al described previously with slight adjust. The description and corresponding references were added in the text.

Results

Comment 3: Describe the major findings of your study in the opening sentence.

Lines 324-325:  This sentence "materials and methods" ?

Response: Thanks for reminding. We moved the sentence to "materials and methods" (2.7 Yeast one hybridization).

Comment 4: Figure 5. B : The picture is not clear

Response: Thanks for the reminding. We adjusted the brightness and contrast of this image and re-labeled it to make it more readable.

Comment 5: All results should be clearly described and logically arranged. References to the respective tables or figures should be in brackets after the statement, e.g., (Table 1) or (Fig. 3A). Use subheadings only when the text is too long and complicated

Response: Thanks for the reminding. We checked the description of results and corrected the above mistakes in paragraph 3.1, 3.2 figure 2 and Table1.

Discussion

Comment 6: The discussion needs to be visited. Appropriate comparisons of results obtained with those found previously should be presented, so I suggest this discussion be re-worded.

Response: Thanks very much for the reviewer’s insightful comment. We re-worded the discussion. We added discussion about the anthocyanin accumulation under low temperature in 18M-245 and comparisons of these results to the similar reports in purple Chinese cabbage seedlings and apple fruit. And we also add further references about the transcription factors BrTT8, BrEGL3, BrTTG1 and BrMYBL2 and discussed in more detail about the mechanism of transcription factors regulating structural genes in anthocyanin biosynthesis.    

Comment 7: Line 403-405:  Can you expand and justify why this is the case?

Response: Anthocyanin accumulation in 18M-245 under low temperature is related to the increase in structural genes expression. It has been reported that the expression of structural genes are regulated by MYB and many other transcription factors in a complicated way. In order to find the main structural gene whose expression change is responsible for anthocyanin accumulation in 18M-245, we determined the expression pattern of BrLDOX, BrDFR and other structural genes in Jiao Erye and 18M-245 under different temperatures. Among all the structural genes tested, only the expression of BjuB014115 was consistent with the anthocyanin accumulation pattern in 18M-245 under low temperature, suggesting that the highly expressed BjuB014115 is responsible for the purple color trait of 18M-245. We add the explain to the text.  

Comment 8: Line 417-422: Extend the conclusion by writing main results of this study. The conclusion has to be improved.

Response: Thanks very much the reviewer’s insightful comment. We extend the conclusion as suggested.

References

Comment 9: Please check all references as several in the main text are old. The author must new cite references at least 75% of the latest reviews (e.g., from 2017-2022).]

Response: Thanks very much for the reminding. We added several latest reviews and studies and corresponding references into the introduction and discussion.

Reviewer 4 Report

In this paper a new line was obtained from the crossing of green and purple cabbages. In this new line was identified a monosomic alien chromosome which carried a LDOX genes responsible for the increased anthocyanin content which is sensitive to different temperature treatments. The regulation of the gene LDOX may follows the same scheme as in Arabidopsis where the authors demonstrated that transcription factors such as members of the MYB family can bind to the cis-elements in the promoter of the gene.

Please, indicate the germination conditions: medium and light.

In the Analysis of anthocyanin content specify the age of the plants the leaves were collected from as you did in 126 line.

In figure 2B I see an increased anthocyanin content more than 10% between 22° and 12° you describe in the lines 252-253. In the figure please specify what the error bars are referred to and the statistic test applied to find significant differences among treatments. Significant differences should be marked with asterisk or letters based on the test used.

Please, in the qRT-PCR section specify the number of technical and biological replicates you used. Again, in the figure 4 please specify what the error bars are referred to and the statistic test applied to find significant differences among treatments/genotypes. Significant differences should be marked with asterisk or letters based on the test used.

To have a clearer view of the pathway of the anthocyanin biosynthesis it would be useful to present the expression pattern of all the genes mentioned in the introduction CHS (chalcone synthase), CHI (chalcone isomerase), F3H (flavanone 3-hydroxylase), F3’H (flavanone 3’-hydroxylase) and FLS (flavonol synthase), UGT (UDP-glucosyltransferase) and AT (acyltransferase) as they did for LDOX and DFR under the three different temperatures. They can be added to the supplemental information.

In the paragraph 3.1.6 please provide further references on the transcription regulation of the anthocyanin biosynthesis genes by MYB and bHLH transcription factors even investigated through different approaches such as ChIP experiment or luciferase assay. This aspect need to be treated in more detail in the discussion reporting further evidences of the action of these transcription factors.  

The discussion should be extended. You don’t mention at all the increasing of anthocyanin accumulation in 18M-245 leaves from 22° to 12°. This topic deserves to be pointed out largely and properly.

Author Response

Dear reviewer:

Thank you very much for giving us the opportunity to revise this manuscript. We appreciate your insightful and constructive comments to improve our manuscript. Below, we address all the comments and questions point-by-point. And of course the text has been changed accordingly, which can also be tracked in the attached revised manuscript. 

In this paper a new line was obtained from the crossing of green and purple cabbages. In this new line was identified a monosomic alien chromosome which carried a LDOX genes responsible for the increased anthocyanin content which is sensitive to different temperature treatments. The regulation of the gene LDOX may follows the same scheme as in Arabidopsis where the authors demonstrated that transcription factors such as members of the MYB family can bind to the cis-elements in the promoter of the gene.

Comment 1: Please, indicate the germination conditions: medium and light.

Response: The seeds were geminated on 1/2 MS medium in dark for 36h. We add these details to the text.

Comment 2: In the Analysis of anthocyanin content specify the age of the plants the leaves were collected from as you did in 126 line.

Response: 6-week-old Chinese cabbage leaves grown under 4, 12 and 22℃ for 4 weeks, respectively were collected for the analysis of anthocyanin content. We added these details to the text.

Comment 3: In figure 2B I see an increased anthocyanin content more than 10% between 22° and 12° you describe in the lines 252-253. In the figure please specify what the error bars are referred to and the statistic test applied to find significant differences among treatments. Significant differences should be marked with asterisk or letters based on the test used.

Response: Sorry for the mistake. We correct the statement as “growth for four weeks at 12℃ increased the anthocyanin content of the purple Chinese cabbage leaves significantly by about 100% compared to 22℃”. The error bars mean ±SE. And we added significant difference marked with asterisk on the figure and legend.

Comment 4: Please, in the qRT-PCR section specify the number of technical and biological replicates you used. Again, in the figure 4 please specify what the error bars are referred to and the statistic test applied to find significant differences among treatments/genotypes. Significant differences should be marked with asterisk or letters based on the test used.

Response: Thank very much for the reviewer’s insightful comment. We did at least three biological replicates of each Q-PCR experiment. We added this statement into the figure legend. And we also added description of error bar and significant difference marked with asterisk to the figure and legend as suggest.

Comment 5: To have a clearer view of the pathway of the anthocyanin biosynthesis it would be useful to present the expression pattern of all the genes mentioned in the introduction CHS (chalcone synthase), CHI (chalcone isomerase), F3H (flavanone 3-hydroxylase), F3’H (flavanone 3’-hydroxylase) and FLS (flavonol synthase), UGT (UDP-glucosyltransferase) and AT (acyltransferase) as they did for LDOX and DFR under the three different temperatures. They can be added to the supplemental information.

Response: Thanks very much for the reviewer’s insightful comment. We determined the expression of the above structural genes in Jiao Erye and 18M-245 grown under three different temperatures using Q-PCR. The results were added to the supplemental data as new Figure S3. Expression of the most genes (excepting FLS2, which is depressed at 4℃ ) were obviously induced in 18M-245 at 4℃, inconsistent with the anthocyanin accumulation pattern of 18M-245 under low temperature. These results suggest that upregulation of BrLDOX is related to anthocyanin accumulation in 18M-245. We added the above description into paragraph 3.4.

Comment 6: In the paragraph 3.1.6 please provide further references on the transcription regulation of the anthocyanin biosynthesis genes by MYB and bHLH transcription factors even investigated through different approaches such as ChIP experiment or luciferase assay. This aspect need to be treated in more detail in the discussion reporting further evidences of the action of these transcription factors.  

Response: Thanks for the reviewer’s insightful comment. We added further references on the anthocyanin biosynthesis regulation by MYB and bHLH transcription factors through yeast one-hybrid and electrophoretic mobility shift analyses (EMSA) into paragraph 3.6. And we revised the corresponding discussion as suggested. 

Comment 7: The discussion should be extended. You don’t mention at all the increasing of anthocyanin accumulation in 18M-245 leaves from 22° to 12°. This topic deserves to be pointed out largely and properly.

Response: Thanks very much for the reviewer’s insightful comment. We have re-write the discussion. Anthocyanin can act as antioxidants in protecting the plant from biotic and abiotic stresses by scavenging ROS. It has been reported that MdMYB2 in apple was able to response to cold stress and enhance the corresponding stress tolerance by promoting anthocyanin accumulation. Thus we predicted that increased accumulation of anthocyanin in 18M-245 is might assist Chinese cabbage in defense to cold stress. We added the above discussion into the “discussion” section.

Round 2

Reviewer 4 Report

The Authors worked on the criticism I suggested and the MS has been improved.

-In the new supplementary material file I don't see the figure S4 with the gene expression.

-In the captions you should correct the sentence: "Data are shown as ±SE" for example "Data are shown as the mean value ±SE".

-When you refer to gene expression the the standard in the field is RT-qPCR.  Please correct it in the text and in the captions. 

Author Response

Dear reviewer,

  Thank you very much for reviewing our manuscript and for your insightful and constructive comments. We appreciate every comment, which has greatly improved our manuscript. A revised manuscript according to your comments is attached. The point to point revision notes are below. The original reviewer’s comments are italicized and our response follows as plain text.

The Authors worked on the criticism I suggested and the MS has been improved.

Comment 1: -In the new supplementary material file I don't see the figure S4 with the gene expression.

Response: Thanks for the reminding. Expressions of other structural genes in anthocyanin biosynthesis in Jiao Erye and 18M-245 grown under different temperatures were showed in the Figure S3. We reinserted the image of Figure S3 and Figure S4.

Comment 2: -In the captions you should correct the sentence: "Data are shown as ±SE" for example "Data are shown as the mean value ±SE".

Response: Thanks very much for the reminding. We corrected the sentence in the legend of Figure 2 (page 12, line 513-514) and Figure 4 (page 13, line 538-539).

Comment 3: -When you refer to gene expression the the standard in the field is RT-qPCR.  Please correct it in the text and in the captions. 

Response: Thanks very much for the reminding. We corrected Q-PCR to RT-qPCR in method 2.5 (page 6, line 269, 275), result 3.4 (page 9, line 398), figure 4 legend (page 13, line 532, 540), and figure S3 legend.
